# Adenoviral-Vectored Multivalent Vaccine Provides Durable Protection Against Influenza B Viruses from Victoria-like and Yamagata-like Lineages

**DOI:** 10.3390/ijms26041538

**Published:** 2025-02-12

**Authors:** Matthew J. Pekarek, Adthakorn Madapong, Joshua Wiggins, Eric A. Weaver

**Affiliations:** 1Nebraska Center for Virology, University of Nebraska-Lincoln, Lincoln, NE 68503, USA; mpekarek2@huskers.unl.edu (M.J.P.); amadapong2@unl.edu (A.M.); jwiggins9@unl.edu (J.W.); 2School of Biological Sciences, University of Nebraska-Lincoln, Lincoln, NE 68503, USA

**Keywords:** influenza B virus (IBV), influenza vaccine, multivalent, Victoria-like, recombinant adenovirus (rAd), cross-lineage, durability

## Abstract

Despite annual vaccines, Influenza B viruses (IBVs) continue to cause severe infections and have a significant impact and burden on the healthcare system. Improving IBV vaccine effectiveness is a key focus, with various strategies under investigation. In this research, we used a computational design to select wildtype sequences for a multivalent viral-vectored vaccine (rAd-Tri-Vic) targeting the Victoria-like (Vic) hemagglutinin (HA) protein. In mouse models, the vaccine induced strong antibody and T cell responses, providing complete protection against both lineage-specific and cross-lineage (Yamagata-like) lethal challenges. The immune responses remained robust for up to six months, which demonstrated sustained protection. These results highlight the potential of HA-targeted multivalent vaccines to enhance the IBV efficacy and address protection against antigenically diverse IBV strains.

## 1. Introduction

The influenza B virus (IBV) is one of the two types of influenza viruses that cause severe disease in humans via seasonal epidemics. IBVs cause nearly one-fourth of all cases during a given season [1]. From the early 1980s to the initial spread of SARS-CoV-2, IBVs circulated as two distinct lineages, known as the Victoria-like (B/Vic) and Yamagata-like (B/Yam) lineages [2,3,4]. The cocirculation of IBV lineages and limited cross-reactive immune responses from seasonal influenza vaccines containing a single IBV lineage [5,6,7,8,9] led to the recommended use of quadrivalent vaccines in the United States, starting in the 2013–2014 influenza season [10]. However, despite both lineages having been incorporated in the quadrivalent seasonal vaccine, the vaccine effectiveness for IBVs remains variable [11]. Therefore, more effective vaccines to better control the IBV are needed.

Various strategies were tested and show promise to improve upon current vaccine standards. Diverting antibody recognition away from antigenically variable sites toward more conserved regions of the IBV hemagglutinin (HA) protein has shown protection through Fc-mediated functions, such as antibody-dependent cellular cytotoxicity (ADCC) [12,13,14]. Others delivered multivalent vaccines targeting multiple proteins [15] to broaden the cross-reactivity of immune responses induced by vaccination. Further studies developed consensus immunogens to limit the antigenic variation from the vaccine immunogen [16,17]. These pre-clinical studies showed protection from broad IBV strains through immune correlation studies and lethal challenge models.

To address strain-specific restrictions observed by traditional vaccines, our group used the epigraph vaccine designer to identify a set of three (trivalent) broadly cross-reactive wildtype HA immunogens. Previous studies that targeted swine and human-origin influenza A viruses (IAVs) using this approach have reported promising rapid, long-lasting, and broadly protective efficacy [18,19,20]. However, a limited sequence population may incorporate uncommon epitopes that disrupt protein structure and structure-targeting immune responses. Therefore, we characterized the immune response to a trivalent (Tri), lineage-specific B/Vic (Vic) HA vaccine formulated using wildtype IBV HA sequences. The wildtype sequences were guided using a computational approach that identified the relevant clusters of previously circulating IBV Victoria-like virus strains. These sequences were cloned into replication-defective recombinant-adenoviral (rAd-Tri-Vic) vectors and administered to naïve mice. The vaccine conferred protection against both lineage-specific and cross-lineage challenges in the murine model at the peak of immunity. Notably, the immune responses were sustained, and the protection remained effective over time. These findings support the potential of lineage-specific B/Vic HA vaccines to provide durable immunity against newly emerging or re-emerging antigenically divergent IBV lineages.

## 2. Results

### 2.1. Vaccine Design and Production

To select wildtype HA sequences that best represent the primary genetic clusters of circulating strains of Victoria-like IBV viruses, we downloaded all unique sequences available on the Influenza Research Database available as of December 2020. The B/Vic lineage sequences were aligned and used as input for the epigraph design algorithm [21]. Once the trivalent epigraph sequences were produced, the sequences were re-aligned onto the unique IBV HA sequence tree and genetically similar wildtype HA sequences (B/Vermont/2/2012, B/Malaysia/2506/2004, B/Victoria/2/1987) were chosen (WT HA) for inclusion in the rAd-Tri-Vic vaccine. Figure 1A shows the phylogenetic relationship between the rAd-Tri-Vic immunogen sequences and the total population of unique B/Victoria sequences. Then, each individual vaccine immunogen sequence was cloned into a replication-defective human adenovirus type 5 (HAdV-5) genome to be delivered as a replication-defective viral-vectored vaccine (rAd-Tri-Vic). The protein expression from the infected HEK293 cells was confirmed by a Western blot analysis, as shown in Figure 1B.

### 2.2. Immediate Protective Immune Responses

To assess the immediate protective immune responses, groups of five mice were vaccinated with a total of 1 × 10^10^ virus particles (vps) of the rAd-Tri-Vic cocktail (3.33 × 10^9^ vps of each immunogen), 600 ng total protein from the 2018–2019 formulation of the Fluzone^®^ Quadrivalent vaccine (Fluzone^®^), or a DPBS sham. Previous studies from our group showed that this group size was sufficient to observe statistically significant differences between experimental and commercial vaccine controls [18,19,20] while limiting the total number of animals used for these studies. The Fluzone^®^ components from this season were B/Colorado/06/2017 as the B/Vic-lineage strain and B/Phuket/3073/2013 as the B/Yam-lineage strain. Three weeks post-vaccination, the mice received a homologous boost vaccine, and two weeks post-boost, the mice were sacrificed for tissue collection. Hemagglutination inhibition (HI) titers are considered the standard of neutralizing antibodies for influenza vaccines. An HI titer of 40 is accepted as a general correlate of protection from human influenza strains [22,23]. The current dogma is that immunity to B-HA induced by inactivated vaccination is limited and does not induce cross-lineage protective immunity [24]. However, the induction of cross-lineage HI titers after using an rAdV-5 delivery vector has not been studied. We performed HI titers to identify the neutralizing antibodies and IFN-γ ELISpot assays to analyze the total antigen-specific T cell responses. Our virus panel included nine B/Vic lineage strains and four B/Yam lineage strains to assess the lineage-specific and cross-lineage HI responses. Three overlapping peptide libraries (two B/Vic and one B/Yam-lineage libraries) were used to assess the lineage-specific and cross-lineage T cell responses following the vaccination. A sequence alignment of the HA proteins used in this study showed the variable regions between the virus strains (Figure 2A). A complete alignment of all proteins used in this study is shown in Appendix A. The phylogenetic relationships between the viruses and vaccines are shown as the percent of identity and similarity (Figure 2B). A phylogenetic tree illustrates the genetic relationships of the HA proteins when analyzed by an uncorrected neighbor-joining phylogram (Figure 2C).

Two weeks post-boost, the rAd-Tri-Vic immunized mice developed protective HI titers against seven of the nine B/Vic strains tested in our panel, with another (FL/15) remaining just below protective levels (Figure 3A). The Fluzone^®^-immunized mice did not reach protective HI titers against any B/Vic strains but did reach protective HI titers against two of the four B/Yam strains assessed (Figure 3A). No cross-lineage HI titers were observed after the rAd-Tri-Vic vaccination. After the restimulation of the harvested splenocytes, we observed the induction of IFN-γ^+^ T cells against both the B/Vic-like overlapping peptide libraries in our rAd-Tri-Vic group (Figure 3B) with no detectable response from the Fluzone^®^ group or sham-immunized mice. Crucially, we observed the induction of IFN-γ^+^ T cells reacting to a B/Yam overlapping peptide library (Figure 3B), suggesting the presence of cross-lineage T cell responses in the absence of neutralizing antibodies. This is consistent with another study that assessed the cross-lineage immune responses to IBV vaccinations [25]. Overall, after two immunizations with rAd-Tri-Vic, we observed the quick induction of a neutralizing antibody and antigen-specific T cell responses to diverse strains of the IBV.

We next performed experimental challenge studies using mouse-adapted IBV strains to determine whether the rAd-Tri-Vic vaccine would protect from lethal infection. Mice were immunized with the same immunization groups and schedule, as described for the immune correlate studies. Two weeks post-boost, the animals were experimentally infected with 20 times the median lethal dose (20 MLD_50_) of either mouse-adapted B/Washington/2/2019 (WA19) (*n* = 5) or B/Malaysia/2506/2004 (Mal04) (*n* = 5). We previously showed that differences in protection can be observed using a stringent lethal challenge with this group size [18,19,20]. The mice were monitored for two weeks post-infection for weight loss and mortality (represented by a loss of 25% of the animals’ body weight at the time of infection).

After the lethal WA/19 challenge, the mice that received the rAd-Tri-Vic vaccine were completely protected from weight loss (Figure 4A) and mortality (Figure 4B). This suggests that the vaccination provided robust protective immunity quickly after vaccination. The Fluzone^®^-vaccinated mice were also completely protected from mortality (Figure 4B); however, they showed signs of weight loss (Figure 4A). Upon further analysis, the mice immunized with rAd-Tri-Vic lost significantly less weight than the mice immunized with Fluzone^®^ (Figure 4C). All sham-vaccinated mice succumbed to infection by eight days post-infection (Figure 4B). Similar trends were observed after the Mal/04 challenge. The mice immunized with rAd-Tri-Vic were completely protected from weight loss (Figure 4D) and mortality (Figure 4E) following the infection with 20 MLD_50_ Mal/04. However, the Fluzone^®^-immunized mice showed signs of weight loss (Figure 4D), but all the mice survived the challenge (Figure 4E). This time, upon further analysis, the difference in weight loss observed between the rAd-Tri-Vic- and Fluzone^®^-immunized mice was not significant (Figure 4F), suggesting that the Fluzone^®^ mice were better protected against this older isolate than the newer WA/19 strain**.** Again, all DPBS sham-vaccinated mice succumbed to infection by eight days post-infection due to weight loss. Overall, these results suggest that rAd-Tri-Vic provided robust protective immunity post-vaccination, and that the immunity induced can protect from a lethal challenge in mice.

While cross-reactive immune responses between B/Vic and B/Yam have been consistently described [26,27,28,29], cross-lineage protection induced by seasonal vaccination is limited [5,7]. Following the same immunization schedule as previously described, groups of 10 mice were challenged with 20 MLD_50_ B/Phuket/3073/2013 (Phu/13), the B/Yam component of the Fluzone^®^ 2018–2019 formulation. Again, the mice were monitored for 14 days post-infection for weight loss and mortality. A larger group size was utilized for this challenge to ensure the reproducibility of the results for this cross-lineage challenge and that additional statistical power was achieved. We observed complete protection from a lethal Phu/13 challenge following vaccination with rAd-Tri-Vic regarding both weight loss (Figure 5A) and survival (Figure 5B), suggesting that the cross-lineage protection afforded by our vaccine was robust. When compared with the Fluzone^®^ group, the rAd-Tri-Vic-vaccinated mice lost significantly less weight despite being a cross-lineage vaccination compared with a matched-strain vaccination (Figure 5C). Again, all DPBS sham-vaccinated mice succumbed to the lethal infection (Figure 5B). Taken together, these results support previous findings that experimental vaccination can induce cross-lineage challenge protection [15]. The results also support that rAd-Tri-Vic holds strong potential to protect from future encounters from lineage-matched and more antigenically diverse challenge strains despite only being designed against the B/Vic lineage.

### 2.3. Duration of Immune Correlates up to Six Months Post-Vaccination

Since an ideal vaccine provides long-term immunity, we tested the duration of the immune responses to rAd-Tri-Vic in the mouse model. This translational study was pursued due to the current circulation of only B/Vic strains and the possible induction of cross-lineage immunity should a new lineage emerge. Groups of 10 mice were vaccinated with 1 × 10^10^ vps of either the rAd-Tri-Vic vaccine cocktail, 600 ng total protein Fluzone^®^, or the DPBS sham vaccine and boosted with the same dose three weeks later. The mice were then housed for an additional 69 days before half (*n* = 5) of the mice were sacrificed for a terminal bleed and spleen harvest to investigate both the humoral and cell-mediated immune responses 90 days post-vaccination. The remaining mice in each group (*n* = 5) were then housed for an additional 90 days before harvest for a total of 180 days post-vaccination prior to tissue collection.

The humoral immune responses were again assessed by the HI titer using the same panel of 13 IBV strains to look for lineage-specific and cross-lineage HI antibody induction. In the D90 group of mice immunized with rAd-Tri-Vic, we observed antibodies from the vaccinated serum, which demonstrated seven out of nine (78%) B/Vic strains at a robust titer (≥40) (Figure 6A). The Fluzone^®^-vaccinated serum did not harbor antibodies with an HI titer of 40 against any of the B/Vic strains tested in the panel at 90 days post-vaccination, only reaching an HI titer of 40 against the matched Phu/13 strain (Figure 6A). The HI titers from the rAd-Tri-Vic vaccinated serum were significantly greater than the Fluzone^®^-vaccinated serum in seven B/Vic strains. The only strain that Fluzone^®^ induced a greater HI antibody response than rAd-Tri-Vic vaccination at day 90 post-vaccination was the matched Phu/13 strain (Figure 6A). No detectable HI antibodies were observed in the DPBS sham-vaccinated animals. At D180, sera from the mice vaccinated with rAd-Tri-Vic maintained an HI titer ≥ 40 in five out of nine (55%) B/Vic strains in the panel (Figure 6B). The Fluzone^®^-vaccinated mice did not reach a serum HI titer of 40 against any of the B/Vic or B/Yam strains tested in the panel (Figure 6B). The differences in HI titers between the rAd-Tri-Vic and Fluzone^®^ groups were significant in six of the B/Vic strains (Figure 6B). When compared with the HI antibody titer results on day 35, we observed that some of the strains that developed low levels of neutralizing antibodies waned over the duration of this study (Figure 3A and Figure 6). However, for the strains that did induce protective levels of neutralizing antibodies quickly after vaccination, the HI titers appeared to be maintained over the course of the durability study. Additionally, the cross-reactive HI antibody responses induced by rAd-Tri-Vic were broad within the B/Vic lineage but did not provide cross-lineage HI responses against the B/Yam strains. These results align with the current understanding that cross-lineage HI antibodies are rare unless developed over time after repeated exposure to IBV HA [30,31], which was not investigated in this study. Taken together, these results demonstrate that the HI antibody responses induced by the rAd-Tri-Vic vaccine were durable for up to 180 days post-vaccination.

Cell-mediated immune responses are known to be important for the clearance of influenza virus once infection has begun [32,33,34]. Studies also highlighted conserved responses to multiple epitopes of IBV that induce cross-reactive T cell responses after vaccination with IBV HA [35,36]. Indeed, our previous work with human H3 trivalent epigraph vaccines showed that T cells were crucial to the protective efficacy in the mouse model [19]. Therefore, the maintenance of antigen-specific T cells from our rAd-Tri-Vic vaccine over time will likely be an important measure of the duration of protection.

To assess the duration of antigen-specific T cell responses to rAd-Tri-Vic, splenocytes were isolated alongside the serum at 90 and 180 days post-vaccination and quantified through an IFN-γ^+^ ELISpot assay. We used the same three overlapping peptide libraries used for the day 35 experiments. At 90 days post-vaccination (Figure 7A), rAd-Tri-Vic-vaccinated mice maintained detectable levels of IFN-γ^+^ T cells that recognized the B/Brisbane/60/2008 (Bris60/08) overlapping peptides. The antigen-specific T cell responses were significantly greater than the Fluzone^®^-immunized group and DPBS sham vaccine, which were nearly undetectable. The T cells that responded to the Mal/04 peptide library (Figure 7A) were lower but still detectable for three out of five rAd-Tri-Vic-vaccinated mice. However, the rAd-Tri-Vic responses were not significantly greater than the Fluzone^®^ or DPBS sham groups. The mice in these control groups all had very little detectable IFN-γ^+^ reactivation despite restimulating with a matched-strain peptide library with one of the three rAd-Tri-Vic immunogen sequences (Figure 1A and Figure 7A, middle). Finally, when splenocytes isolated 90 days post-vaccination were restimulated with B/Nanchang/12/1998 (Nan/98) HA peptides (Figure 7A), the rAd-Tri-Vic splenocytes were only detected at very low levels, with no differences between the groups observed. Using the same overlapping peptide libraries, we assessed the antigen-specific T cell responses of splenocytes isolated 180 days post-vaccination (Figure 7B). Similar trends were shown with the rAd-Tri-Vic mice that responded to all three peptide libraries, while the Fluzone^®^-immunized mice remained nearly undetectable. After restimulation with the Bris60/08 peptide library, the rAd-Tri-Vic mice showed significantly greater levels of IFN-γ production than the Fluzone^®^-immunized mice (Figure 7B). Unlike at day 90, the rAd-Tri-Vic group T cell responses were significantly greater than those with Fluzone^®^ when restimulated with Mal/04 HA peptides at 180 days post-vaccination. Finally, the rAd-Tri-Vic mice showed significantly greater levels of IFN-γ production after restimulation with Nan/98 HA peptides than the Fluzone^®^-immunized splenocytes (Figure 7B). These results suggest that vaccination with rAd-Tri-Vic induces T cells that remain in circulation up to 180 days post-vaccination and reactivate upon subsequent HA peptide exposure. Additionally, these T cells reacted to a cross-lineage peptide library, further supporting the potential for cross-lineage T cell responses as a target for IBV vaccination [15,36]. Overall, the samples obtained from mice 90 and 180 days post-vaccination suggest that rAd-Tri-Vic vaccination can induce durable, HA-specific immune responses up to six months post-vaccination.

### 2.4. Duration of Protective Immunity Six Months Post-Vaccination

After showing that we could detect immune responses that were maintained up to six months post-vaccination, we also wanted to assess whether these immune responses protected from a challenge. The vaccinated mice (*n* = 8–10 per group) that had been housed for 180 days post-vaccination were infected with the same 20 MLD_50_ dose that was previously used for the challenge studies and monitored for 14 days post-infection. Differences in the animal number between groups were caused by the loss of individual animals due to circumstances unrelated to this study during the six-month duration. Following the challenge with WA/19, the mice vaccinated with rAd-Tri-Vic showed signs of slight weight loss (~6% of initial weight) on day four post-infection (Figure 8A) but were completely protected from mortality (Figure 8B). The Fluzone^®^-immunized mice showed signs of more severe weight loss post-infection (~10.5% weight loss at day five post-infection) (Figure 8A) but were also protected from mortality (Figure 8B). When comparing the weight loss curves of the animals from each group (Figure 8C), the differences in weight loss between the Fluzone^®^- and rAd-Tri-Vic-immunized mice were near statistical significance but did not achieve statistical significance. Unlike the previous challenge studies, two mice in the DPBS sham group survived the infection (Figure 8B), indicating that the lethal dose titer from 8–10-week-old mice was not maintained in mice that were 7–8 months of age. However, all vaccinated mice were significantly more protected from mortality than the DPBS sham-vaccinated mice, indicating that rAd-Tri-Vic vaccination can induce significant protection from a B/Vic challenge up to six months post-vaccination.

To assess the cross-lineage protection, we again infected the mice that had been housed for 180 days post-vaccination with the same 20 MLD_50_ dose from the previous challenges conducted with Phu/13. The rAd-Tri-Vic-vaccinated mice showed signs of moderate weight loss early after infection (~7.5% weight loss by day four post-infection) (Figure 8D) but maintained their weight at this level and all survived the infection (Figure 8E). The Fluzone^®^-immunized mice reached peak weight loss slightly later post-challenge (~6% weight loss on day six post-infection) (Figure 8D) and regained some weight by day 14 to all survive the challenge (Figure 8E). When comparing the differences in weight loss between the groups (Figure 8F), no significant differences were observed between the Fluzone^®^-immunized mice and the rAd-Tri-Vic vaccine group. As with the WA/19 D180 challenge, the lethal dose used for the infection that was previously titered in 8–10-week old mice only reached a mortality of 80% in the sham-vaccinated mice at 7–8 months of age (Figure 8E), but the vaccinated mice were significantly more likely to survive the infection than the unvaccinated group. These results suggest that the protection from the rAd-Tri-Vic vaccination observed at peak immunity was maintained up to six months post-vaccination. Furthermore, this protection was maintained against a lineage-matched and cross-lineage challenge, and the cross-lineage protection observed was not significantly worse than a matched-strain licensed influenza vaccine. Altogether, this multivalent vaccine design and delivery platform shows tremendous potential to improve upon protection afforded by the currently licensed vaccines.

## 3. Discussion

One of the primary objectives for a universal influenza vaccine is to induce durable immunity longer than a single influenza season to limit the annual updates required [37,38]. Our previous work with the epigraph vaccine designs targeting influenza A viruses suggested that this strategy can produce broad cross-reactive immune responses within the target lineage [18,19]. However, the more limited sequence diversity among IBVs may limit the effectiveness of the computational design platform to produce immunogens that function and maintain structural conformation similar to wildtype HAs. Due to this concern and the synthetic nature of the epigraph immunogens, we sought to utilize this approach to identify existing wildtype immunogens that may best represent the overall genetic diversity of the IBV while maintaining the native conformational structures that wildtype HA proteins present. In this paper, we report on the efficacy of computationally guided wildtype HA immunogen selection to deliver a multivalent B/Vic vaccine that maximizes possible cross-reactive immunity.

Our initial studies first demonstrated that rAd-Tri-Vic vaccination led to the early detection of receptor-blocking HI antibodies and antigen-specific T cell responses after two immunizations. Furthermore, this immunization provided robust protection from lethal B/Vic challenge, outperforming commercially available immunization when comparing the weight loss two weeks post-immunization. We also observed that our rAd-Tri-Vic vaccine induced strong protection against a cross-lineage B/Yam challenge. Despite the current lack of circulation of B/Yam viruses [3,4], this robust cross-lineage protection could still be valuable should a B/Yam strain re-emerge, or a new B/Yam-like lineage splits off the currently circulating B/Vic lineage. This cross-lineage protection was also described by other groups investigating multivalent IBV vaccines [15] and may be able to serve as a standard of cross-protective immunity in future IBV vaccine pre-clinical studies. Further studies, including defining IgG subclass levels post-vaccination, will help to identify the contribution to the protection of non-neutralizing antibodies induced quickly after rAd-Tri-Vic vaccination. Immune responses at the mucosal sites could also be contributing to the protective efficacy observed in our results. However, a recent study that compared intranasal and intramuscular delivery showed decreased mucosal immune responses from a rAd-vectored vaccine delivered intramuscularly compared with intranasally, suggesting that protection was more likely mediated through a systemic mechanism of protection [39].

While immediate protection is important, effective vaccines should also induce durable immunity. To assess the duration of immunity, we collected sera and spleens at 90 and 180 days post-vaccination to better understand how the immune responses changed over time. When looking at the HI antibody responses, we saw sustained protective titers on both days 90 and 180 post-vaccination in the most genetically similar strains to our rAd-Tri-Vic vaccine. The commercial comparator included in this study only maintained moderate responses against the matched strain or similar strains to the component HAs with little cross-reactivity. We did observe an interesting trend in the antibody responses over time, with the antibodies on day 35 not having reached a peak titer, as had been expected. Rather, the antibody responses for certain strains continued to increase between days 35 and 90. This may be a sign of further affinity maturation taking place between days 35 and 90 for strains more closely related to the vaccine immunogens [40]. However, a more dedicated investigation would be necessary to confirm this hypothesis. For IFN-γ^+^ T cells, we observed similar results. The rAd-Tri-Vic vaccination led to long-lasting T cell responses that were detectable even up to 180 days post-vaccination. In contrast, animals immunized with the commercial comparator vaccine had nearly undetectable IFN-γ^+^ T cells. Other reports of experimental influenza vaccines tested in mouse models targeting influenza HAs also showed a similar maintenance of HI titers to matched strains from 5 to 8 months [41,42,43]. Another study identified T cell responses elicited by a HAdV-5-vectored nucleoprotein vaccine that were also detectable via ELISpot up to 12 months post-vaccination [44]. Therefore, our results align with the current understanding about the duration of immunity to influenza vaccines and further show the capabilities of directly targeting the IBV in a vaccine formulation. Deeper investigation into other classes of T cell responses, including the effectiveness of inducing T follicular helper (Tfh) cells, could shed additional insight to explain the duration of antibody responses observed after vaccination [45,46].

To further analyze the functionality of our vaccine-induced immunity, we challenged mice to determine whether protection was maintained over time. Once again, we observed complete protection from a WA/19 challenge in the vaccinated groups but no longer saw significant differences between the rAd-Tri-Vic group and the commercial comparator. However, the challenge dose that was used, having previously been titered for lethality in 8–10-week-old mice, was no longer completely lethal in the mice that were 7–8 months of age. Another study that investigated flu vaccine effectiveness over time also reported a lethal dose challenge that was not 100% lethal in the older mice [47], so this could have contributed to the lack of differences between groups if the challenge was not stringent enough. As the vaccine design and delivery continue to be optimized, a more stringent lethal challenge will be necessary to completely understand the possible differences in protection over time. Despite this, all vaccinated animals were still significantly protected from mortality compared with the sham vaccine controls. These results were also supported by the complete protection from the Phu/13 challenge. This challenge at 180 days post-vaccination showed that the cross-lineage protection observed at peak immune responses was also maintained up to six months post-vaccination. While cross-protection across lineages was previously shown in experimental vaccines [14,15], this study provided evidence that cross-lineage immunity induced through vaccination could be maintained over time without the need for boosting immunity with the other lineage.

While these results were overwhelmingly supportive of the potential for a B/Vic HA vaccine design, one result of note was the limited HI antibody response against more contemporary strains, specifically from the V1A.3 clade and further subclades. Originally detected in the 2018–2019 North American flu season [48,49], this clade of B/Vic viruses rapidly emerged and began to dominate the global IBV circulation in the following years [50,51,52,53]. Both B/Washington/2/2019 and B/New Hampshire/1/2021 from our virus panel belong to this clade [54,55], and while we detected very little HI antibody, the results from our challenges with B/Washington/2/2019 suggest that the rAd-Tri-Vic vaccination still resulted in protective immunity. The importance of non-neutralizing, Fc-mediated antibody functions, such as ADCC, toward protection from the IBV was previously described [35,56,57]. Furthermore, this non-HI activity was shown to be durable and maintain protection in other studies that characterized the duration of immunity with IBV vaccination [14]. However, additional experiments that assess these potential non-neutralizing protective mechanisms will be necessary to determine their role in protection. This has also gained more significance due to the lack of circulating B/Yam IBVs since the initial response to the COVID-19 pandemic [3,4]. Our results suggest that Fluzone^®^ struggles to induce cross-reactive immune responses within the B/Vic lineage, underscoring the need to develop more effective B/Vic vaccines. Fully understanding the most effective ways of limiting the spread of B/Vic strains by vaccination could lead to a dramatic decline in the human seasonal influenza burden.

This paper details the development of a lineage-specific, multivalent B/Vic vaccine that elicited protective immune responses. Antibody and cell-mediated immune correlates, along with lethal challenge studies, demonstrated broad cross-lineage protection and durability lasting up to six months post-vaccination. Further investigation into the mechanisms underlying this protection will enhance the understanding of how this vaccine design achieves such efficacy. Future translational studies, including evaluating immune responses in the context of pre-existing IBV HA immunity, will provide critical insights into the potential application of multivalent vaccines for preventing IBV infections.

## 4. Materials and Methods

### 4.1. Ethics Statement

The protocols and procedures used in this study involving animals were approved by the Institutional Animal Care and Use Committee of the University of Nebraska–Lincoln under protocol #2158 (approved in November 2021). All experiments followed the guidelines put forth by the Animal Welfare Act, Public Health Service Animal Welfare Policy, and the National Institutes of Health Guide for the Care and Use of Laboratory Animals. The animals were housed under the Association for Assessment and Accreditation of Laboratory Animal Care (AAALAC)-certified facility guidelines at the University of Nebraska–Lincoln Life Sciences Annex. The procedures not involving animals were approved by the Institutional Biosafety Committee of the University of Nebraska–Lincoln (approved November 2015). All the sequences used in this report are publicly available on the Influenza Research Database (https://www.ncbi.nlm.nih.gov/genomes/FLU/Database/nph-select.cgi?go=database (accessed on 11 December 2020)).

### 4.2. Influenza Viruses

The following viruses were provided by the Biodefense and Emerging Infections Research Resources Repository (BEI) of the National Institute of Allergy and Infectious Diseases: B/Malaysia/2506/2004 (Mal/04, NR-12280), B/New York/1056/2003 (NY/03, NR-4861), and B/Florida/4/2006 (FL/06, NR-9696). The following viruses were provided by the International Reagent Resource (IRR) of the Centers for Disease Control and Prevention: B/New Hampshire/1/2021 (NH/21, FR-1796), B/Washington/2/2019 (WA/19, FR-1709), B/Hawaii/1/2018 (HI/18, FR-1661), B/Florida/78/2015 (FL/15, FR-1538), B/Nevada/3/2011 (NV/11, FR-1028), B/Brisbane/33/2008 (Bris33/08, FR-340), B/Victoria/304/2006 (Vic/06, FR-20), B/Oklahoma/10/2018 (OK/18, FR-1660), B/Wisconsin/10/2016 (WI/16, FR-1663), and B/Phuket/3073/2013 (Phu/13, FR-1364). Viruses were expanded for this study using 9–11-day-old specific-pathogen-free (SPF) embryonated chicken eggs (Charles River Laboratories) at 33–35 °C for 48–72 h or serial passage in a Madin–Darby canine kidney (MDCK)-cell culture at 33 °C for 48–72 h. Cell debris was removed from the chorioallantoic fluid or cell culture supernatant by centrifugation at 300× *g* for 10 min at 4 °C. Viruses with a concentration of at least 64 hemagglutination units (HAU) were aliquoted and stored at −80 °C. Lethal mouse-adapted challenge strains were serially passaged, as previously described [58].

### 4.3. Vaccine Sequence Selection and Production

To select the lineage-specific rAd-Tri-Vic vaccine sequences, B-HA sequences were downloaded from the Influenza Research Database using the following parameters: protein sequence, HA protein, influenza type B, complete sequences only, remove duplicate sequences, exclude lab strains, and sequences up to 11 December 2020. This search generated 3516 protein sequences. The sequences were aligned in Geneious (v.11.0.5) using the ClustalW plugin with the following parameters: BLOSUM62 cost matrix, open cost: 10, and gap extend cost: 0.1. A phylogenetic tree was created using the RAxML blackbox plugin using the CIPRES computer with the following parameters: protein sequence, JTT matrix, 72 h, and 1000 bootstraps. Lineage-specific sequences were identified on the phylogenetic tree using the sequences farther outside the tree center from the lineage reference strain B/Victoria/2/1987, which yielded 1673 B/Vic HA sequences. These sequences were aligned using the ClustalW parameters described above. This alignment was then used as the input population for the epigraph vaccine design algorithm [21]. The computational immunogen sequences were screened using BlastP to identify 3 genetically similar wildtype protein sequences (B/Vermont/2/2012, B/Malaysia/2506/2004, and B/Victoria/2/1987). A 6× histidine tag was added to the C-terminal end of all three selected wildtype sequences and synthetically produced by GeneArt Gene Synthesis (Thermo-Fisher Scientific, Waltham, MA, USA) in the pFastBac-1 plasmid. The individual B/Vic HA genes were cloned into a replication-defective human adenovirus serotype 5 (HAdV-5) backbone using the AdEasy Adenoviral Vector System (Agilent Technologies, Santa Clara, CA, USA). Successful incorporation into the HAdV-5 genome was confirmed through restriction digest cloning and plasmid sequencing (Eurofins Genomics, Louisville, KY, USA). Vaccine virus particles were produced after the transfection of HEK293 cells with one of the HAdV-5 genomes containing a transgene and harvest of cells after plaque formation. Sequential amplification of the virus up to a 10-layer CellSTACK^®^ (Corning, NY, USA) was completed before virus particles were purified through CsCl gradient ultracentrifugation purification. Vaccine stocks were de-salted with Econo-Pac 10G Desalting Columns (Bio-Rad, Santa Clara, CA, USA) before storage at −80 °C. The total number of virus particles was calculated using the OD_260_ measured by a NanoDrop Lite spectrophotometer (Thermo-Fisher Scientific), and the total particle–infectious unit ratios were calculated after the Adeno-X Rapid Titer Kit (Takara Bio Company, San Jose, CA, USA) analysis per the manufacturer’s instructions.

### 4.4. Western Blot

Western blotting was used to detect the protein expression from the rAd-Tri-Vic vectors. Confluent HEK293 cells in a 12-well plate were infected with 100 virus particles/cell of HAdV-5-vectored individual HA genes for 24 h. The cells were harvested and resuspended in 2× Laemmli buffer (Bio-Rad) with 2-mercaptoethanol. The cells were then incubated at 100 °C for 10 min to ensure cell lysis and protein denaturation. The cell lysates were passed one time through a Qiashredder (Qiagen, Hilden, Germany) before resolution on a 12.5% SDS-PAGE. Proteins were transferred from the gel onto a nitrocellulose membrane via the sandwich method. Membranes were blocked for one hour with 5% milk in tris-buffered saline plus Tween 20 (TBST). To detect the HA proteins, the membrane was incubated at 4 °C overnight with polyclonal goat anti-B/Hong Kong/8/1973 antiserum (BEI, NR-3165). Mouse anti-GAPDH-HRP conjugate (Santa Cruz Biotechnology, Dallas, TX, USA, sc-47724) was used as a cellular loading control. After overnight incubation, the membrane was washed five times in TBST before adding donkey anti-goat IgG-HRP conjugate (Millipore Sigma, Burlington, MA, USA) for one hour at room temperature as a secondary antibody. The membrane was washed five times with TBST before developing for five minutes using the SuperSignal West Pico Chemiluminescent substrate (Thermo Scientific). The membrane was imaged using a ChemiDoc MP Imaging System (Bio-Rad).

### 4.5. Vaccinations and Tissue Collections

Six- to eight-week-old female BALB/c mice (Jackson Laboratory, Bar Harbor, ME, USA) were allocated into three groups: DPBS, Fluzone^®^, and rAd-Tri-Vic. All mice were immunized intramuscularly (i.m.) into each quadricep muscle in two 25 µL injections for a total volume of 50 µL on days 0 and 21. The DPBS sham control mice received 50 µL of DPBS at each immunization. The Fluzone^®^ mice received 600 ng of total protein at each immunization. The rAd-Tri-Vic mice were immunized intramuscularly (i.m.) with 1 × 10^10^ total virus particles (vps) of the HAdV-5-vectored rAd-Tri-Vic vaccine cocktail (3.33 × 10^9^ vps encoding B/Vermont/2/2012 HA, 3.33 × 10^9^ vps encoding B/Malaysia/2506/2004 HA, and 3.33 × 10^9^ vps encoding B/Victoria/2/1987 HA). The vaccine doses used for optimal immunity were chosen based on previously described studies [59]. For the immune correlate experiments, serum was obtained by a cardiac puncture and separated from the whole blood with BD Microtainer^®^ Chemistry Tubes (Becton-Dickinson, Franklin Lakes, NJ, USA). Splenocytes were collected through a 40 µm Nylon Mesh cell strainer (FisherBrand, Canton, MA, USA) to create single-cell suspensions of the harvested spleen. Then, the cells were treated with an ammonium–chloride–potassium (ACK) lysis buffer to lyse red blood cells. The processed splenocytes were suspended in a mixture of 90% v/v fetal bovine serum (FBS) and 10% v/v DMSO (bioWorld, Dublin, OH, USA) and stored in a liquid nitrogen vapor phase. All vaccinations were conducted under isoflurane anesthesia.

### 4.6. Hemagglutination Inhibition Assays

The serum was treated with receptor-destroying enzyme (RDE; Denka Seiken, Bibai, Japan) at a 1:3 serum–RDE ratio at 37 °C overnight. After incubation, the samples were heat-inactivated at 56 °C for 30 min before dilution to a final 1:10 serum–diluent ratio using DPBS. For the hemagglutination inhibition (HI) assay, the 1:10 serum dilution was added to a 96-well V-bottom plate, then serially diluted two-fold. Then, 8 HAU of the respective virus stock described above was added to all wells and incubated with the serum for one hour at room temperature. To determine the HI titer, 50 µL of 0.5% rooster red blood cells (Lampire Biological Laboratories, Pipersville, PA, USA) were added to each well and incubated for 30–45 min at room temperature before reading hemagglutination patterns via teardrop formation after plate elevation.

### 4.7. Enzyme-Linked Immunospot (ELISpot) Assay

IFN-γ ELISpot assays were conducted using overlapping peptide arrays for B/Brisbane/60/2008 HA (Bris60/08; BEI, NR-19247), B/Malaysia/2506/2004 HA (Mal/04; BEI, NR-18967), and B/Nanchang/12/98 HA (Nan/98; BEI, NR-2605) for restimulation. Peptides were pooled to a final concentration of 5 µg/mL prior to use. Ninety-six-well hydrophobic Immobilon-P Membrane (Millipore) plates were coated with 250 ng anti-mouse IFN-γ monoclonal antibody AN18 (Mabtech, Nacka Strand, Sweden) overnight. The next day, the wells were blocked with RPMI-1640 supplemented with 5% v/v FBS and 1% penicillin/streptomycin (P/S) at 37 °C with 5% CO_2_ for two hours. Pooled peptides (50 µL) were added to experimental wells, with 50 µL concanavalin A (5 µg/mL) (Sigma) added to positive control wells, and 50 µL RPMI-1640 with 5% FBS and 1% P/S added to unstimulated control wells. The splenocytes (2.5 × 10^5^ cells/mL) from individual mice were added to all the wells in duplicate and incubated at 37 °C with 5% CO_2_ for 18 h. The plates were then washed six times with DPBS with Tween 20 (DPBS-T) before adding 50 µL biotinylated anti-mouse IFN-γ monoclonal antibody R4-6A2 (Mabtech) at a 1 µg/mL concentration. The plates were incubated at room temperature for one hour, then washed six times with DPBS-T. Streptavidin–alkaline phosphatase (1:1000 dilution; Mabtech) was added to all wells and incubated at room temperature for 45 min. Following six more DPBS-T washes, the plates were developed with 100 µL BCIP/NBT (MOSS Bio, Franklin Park, IL, USA) alkaline phosphatase substrate and incubated at room temperature until spots began to appear in the concanavalin A control wells. Development was stopped by washing repeatedly with de-ionized water. The developed plates were dried and stored in the dark for at least 18 h. Spots were counted using an automatic count function ELISpot plate reader (CTL Cellular Technology, Shaker Heights, OH, USA), and the data are reported in spot-forming units per million splenocytes (SFU/10^6^ splenocytes).

### 4.8. Mouse Challenges

The mice were separated into groups as described above and vaccinated on days 0 and 21. The mice were infected either two weeks post-boost immunization, on day 35, for immediate protective immune response challenges, or on day 180 for durability challenges. The mice were infected intranasally (i.n.) with 20 times the median lethal dose (20 MLD_50_) in 10 µL through each nare, with a mouse-adapted WA/19, Mal/04, or Phu/13 challenge strain. The lethal dose was titered using 8–10-week-old female BALB/c mice using the Reed–Muench method [60]. The virus was delivered under ketamine/xylazine anesthesia. The mice were monitored for 14 days post-infection for morbidity and mortality. The animals that lost at least 25% of their initial body weight were humanely euthanized as an endpoint substitution for death.

### 4.9. Statistical Analysis

All statistical analyses described were completed using GraphPad Prism software v.10.0.

## 5. Conclusions

Influenza B viruses continue to circulate in humans, and current vaccination methods lead to suboptimal protection from infections. Our group previously showed promise in developing computationally designed vaccines to prevent influenza infection in animal models. In this study, we showed that the computational design strategies can be modified to design multivalent wildtype vaccines that lead to protective immunity that is maintained up to six months post-vaccination. These results further highlight the promise of targeting IBVs directly to improve the overall influenza vaccine effectiveness.

## Figures and Tables

**Figure 1 ijms-26-01538-f001:**
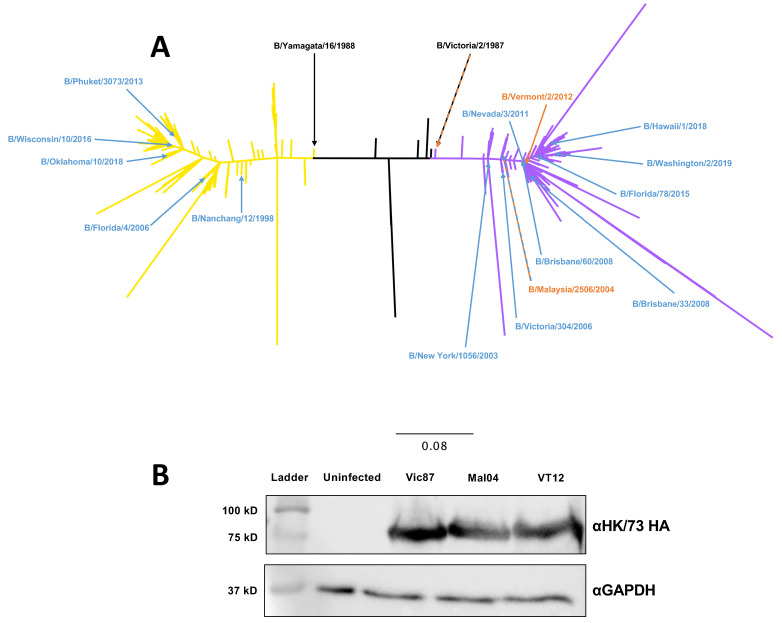
Phylogenetic relationship and protein expression. (**A**) A maximum-likelihood phylogenetic tree showing the genetic relationships of all the unique B-HA sequences that were available in the Influenza Research Database as of 11 December 2020. The reference strains are denoted with black arrows. The analyzed sequences identified as B/Yam-lineage are colored yellow, while the B/Vic-lineage strains are violet. The strains isolated prior to the emergence of the genetic lineages are colored black. The locations of the rAd-Tri-Vic immunogen sequences are denoted with orange arrows and the HA sequences from the strains used in this study are denoted with blue arrows. (**B**) Western blot showing the detection of B-HA expression from HAdV-5-infected HEK293 cells expressing the individual immunogen indicated above using antiserum derived against the B/Hong Kong/8/1973 HA protein. The GAPDH expression is shown as the loading control. A whole blot image is shown in Appendix A.

**Figure 2 ijms-26-01538-f002:**
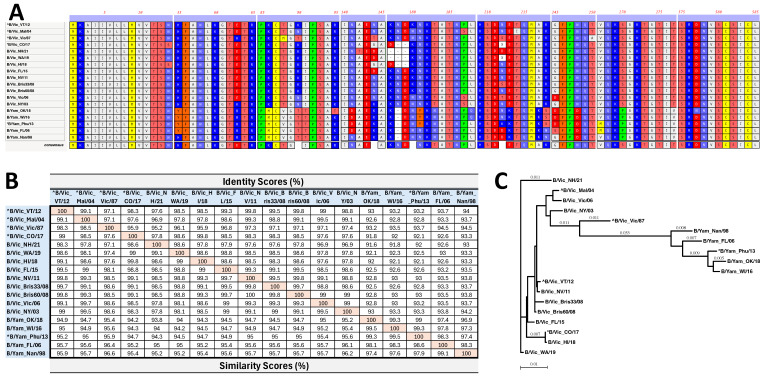
Sequence alignment of the vaccine strains delivered in this study and the strains utilized in the HI titer virus panel and overlapping peptide libraries for IFN-γ^+^ ELISpot assays. rAd-Tri-Vic vaccine strains are denoted by a ^, and strains included in the 2018–2019 Fluzone^®^ formulation used are denoted by a *. The alignment shows the regions with significant diversity and was created using MacVector v18.6.4 (**A**). The percent identity and similarity analysis are shown and indicate the genetic relationship between the proteins used in this study (**B**). Phylogenetic analyses of the HA proteins used were performed using an uncorrected neighbor-joining analysis (**C**). All analyses were performed and represented using MacVector v18.6.4 software.

**Figure 3 ijms-26-01538-f003:**
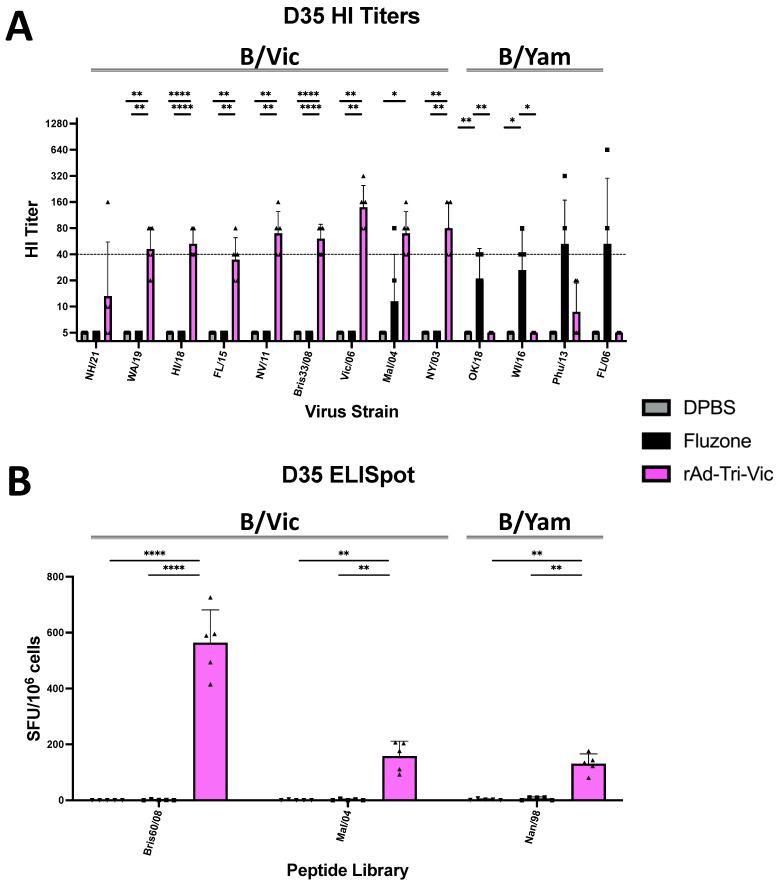
Immune correlate assays two weeks post-boost immunization. (**A**) Hemagglutination inhibition (HI) titers conducted using a total of 13 IBV strains, including 9 B/Vic strains and 4 B/Yam strains. Titers represent the inverse of the highest dilution that prevented the agglutination assessed using the teardrop method. The horizontal dashed line at a titer of 40 represents the threshold accepted by the field as a protective HI titer. (**B**) IFN-γ ELISpot assay results obtained after restimulating harvested splenocytes with one of three overlapping peptide libraries encoding the full-length IBV HA protein. The results are shown as spot-forming units per million splenocytes. A statistical analysis was conducted using one-way ANOVA on each individual strain to determine the significant differences between vaccine groups (* *p* < 0.05, ** *p* < 0.01, **** *p* < 0.0001). Data points from individual animals are denoted by the dotted shapes in each panel.

**Figure 4 ijms-26-01538-f004:**
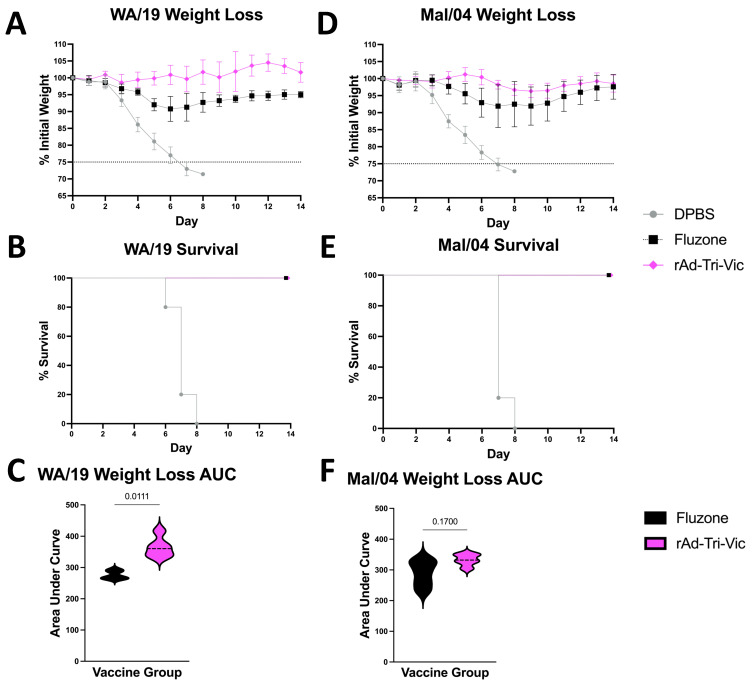
B/Vic lineage-specific challenges two weeks post-boost. The mice were vaccinated on days 0 and 21, and on day 35, infected with 20 MLD_50_ B/Washington/2/2019 (**A**–**C**) or B/Malaysia/2506/2004 (**D**–**F**). The mice were monitored for 14 days post-infection for weight loss (**A**,**D**) and survival (**B**,**E**). The differences in weight loss between the groups (**C**,**F**) were calculated by an area under the curve analysis and analyzed via paired *t*-tests. A 25% weight loss threshold was used for humane euthanasia. The individual mouse weight curves used to calculate the AUC values are in Appendix A.

**Figure 5 ijms-26-01538-f005:**
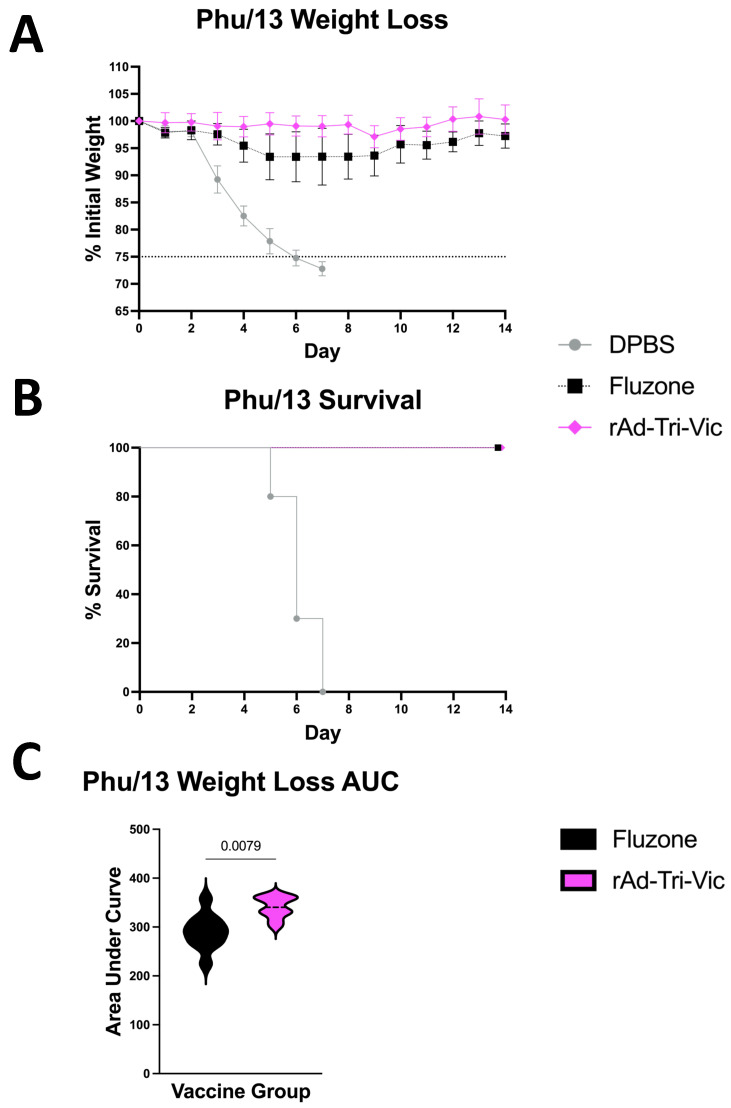
B/Yam cross-lineage challenge two weeks post-boost. The mice were vaccinated on days 0 and 21, and on day 35, infected with 20 MLD_50_ B/Phuket/3073/2013. The mice were monitored for 14 days post-infection for weight loss (**A**) and survival (**B**). The differences in weight loss between the groups (**C**) were calculated by area under the curve analyses and analyzed via paired *t*-tests. A 25% weight loss threshold was used for humane euthanasia. The individual mouse weight curves used to calculate the AUC values are in Appendix A.

**Figure 6 ijms-26-01538-f006:**
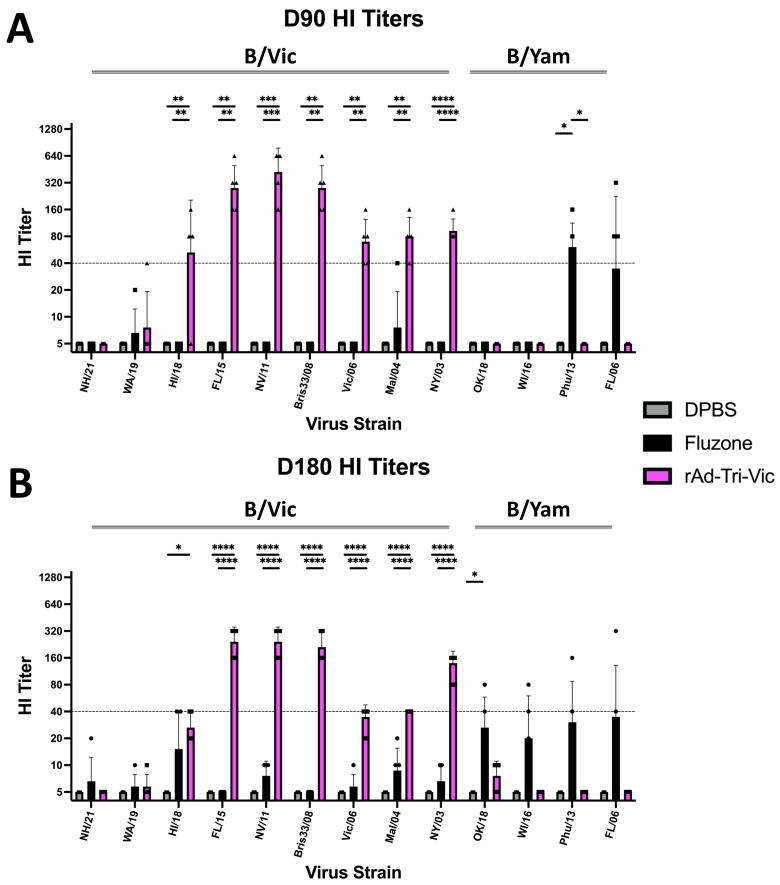
The mice were vaccinated on day 0 and day 21 with either the DPBS sham, Fluzone^®^ commercial comparator, or rAd-Tri-Vic vaccine. Then, the mice were sacrificed on day 90 (**A**) or day 180 (**B**) for a serum harvest via a terminal bleed, and the serum samples were used to quantify hemagglutination inhibition activity against the same panel of 9 B/Vic strains and 4 B/Yam strains used at two weeks post-boost immunization. A statistical analysis was conducted using one-way ANOVA on each individual strain to determine the significant differences between the vaccine groups (* *p* < 0.05, ** *p* < 0.01, *** *p* < 0.001, **** *p* < 0.0001). The dashed line at an HI titer of 40 represents the protective threshold for immunity generally accepted for human influenza viruses.

**Figure 7 ijms-26-01538-f007:**
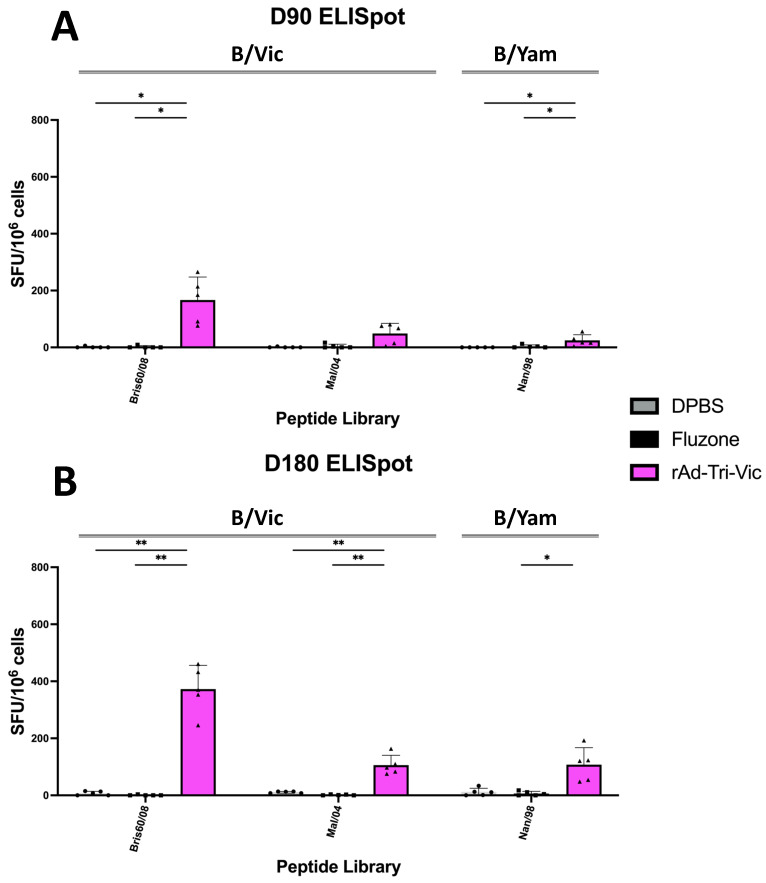
The mice were vaccinated on day 0 and day 21 with either the DPBS sham, Fluzone^®^ commercial comparator, or rAd-Tri-Vic. Then, the mice were sacrificed on day 90 (**A**) or day 180 (**B**) for a spleen harvest to quantify the antigen-specific IFN-γ^+^ T cells that reacted to overlapping peptide library restimulation. The peptide libraries that encoded the full-length HA sequence of 2 B/Vic strains and 1 B/Yam strain were used. The number of spots counted on individual plates were normalized by the cell number, and the results are reported as spot-forming units per million splenocytes (SFU/10^6^ cells). A statistical analysis was conducted using one-way ANOVA on each individual strain to determine the significant differences between the vaccine groups (* *p* < 0.05, ** *p* < 0.01).

**Figure 8 ijms-26-01538-f008:**
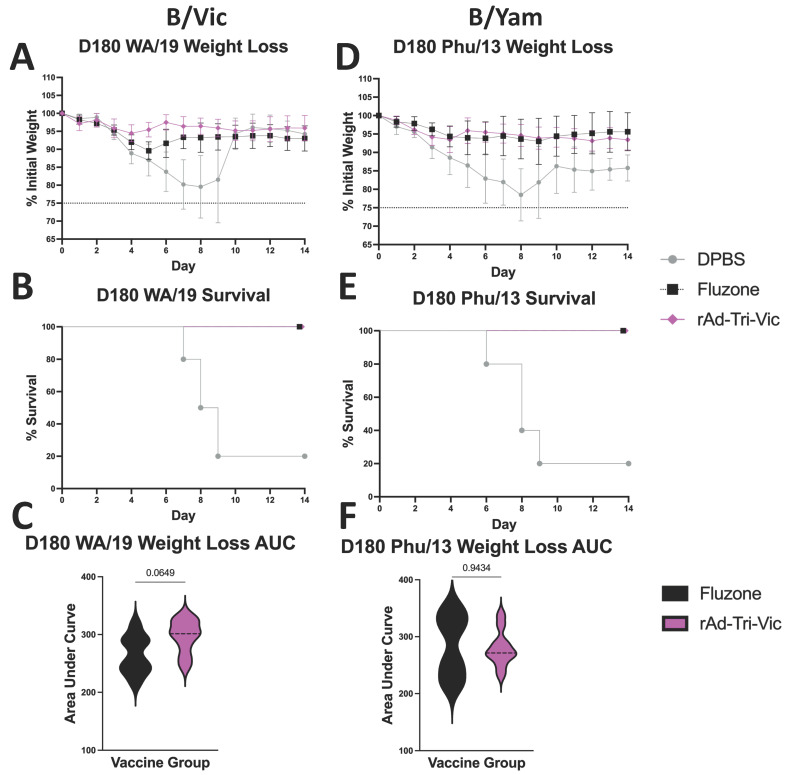
The mice were vaccinated on day 0 and day 21 with either the DPBS sham, Fluzone^®^ commercial comparator, or rAd-Tri-Vic. On day 180, the mice were infected with 20 MLD_50_ B/Washington/2/2019 (**A**–**C**) or B/Phuket/3073/2013 (**D**–**F**) and monitored for 14 days post-infection for weight loss (**A**,**D**) and survival (**B**,**E**) after a challenge. The differences in weight loss between the groups (**C**,**F**) were calculated by an area under the curve analysis and analyzed via paired *t*-tests. A 25% weight loss threshold was used for humane euthanasia. The individual mouse weight curves used to calculate the AUC values are in Appendix A.

## Data Availability

All data relevant to this study are found in the main figures or Appendix A.

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
