# Peer review of "Adenoviral-Vectored Multivalent Vaccine Provides Durable Protection Against Influenza B Viruses from Victoria-like and Yamagata-like Lineages"

_ijms, 2025, doi:10.3390/ijms26041538_

Round 1

Reviewer 1 Report

Comments and Suggestions for Authors

In this manuscript, Pekarek et al, addressed the “Adenoviral-vectored Multivalent Vaccine Provides Durable Protection Against Influenza B Virus ”. Having examined the manuscript, I note that though it discusses interesting observations, to be considered for MDPI International Journal of Molecular Sciences , the following are some of the comments that the authors might find useful for future submission. This manuscript is well-structured and delivers insightful information regarding novel vaccine strategy for controlling Influenza B Viruses. This type of studies are extremely  valuable for the scientific community at a global level.

Reviewer Comments

1.     The title appropriately reflects the content of the manuscript. However, the manuscript addresses critical challenges in influenza B vaccine development, comprising Victoria lineage and shows cross-protection against Yamagata lineage. Highlighting the vaccine's cross-protective abilities in the manuscript title could increase its attractiveness and highlight the broader significance of the findings.

2.     If feasible, I suggest authors to include serum antibody responses such as IgG subclasses (IgG1, IgG2a, IgG2b, IgG3) provide critical information regarding Th1 and Th2 anti influenza immune responses

3.     If feasible, I suggest authors, to quantify mucosal IgA responses in nasal washes and other mucosal secretions to provide valuable insights to mucosal immunity even though the vaccine administered intramuscularly.

4.     The authors evaluated IFN-gamma T cell responses which is crucial for understanding cellular immunity. However, measuring additional T cell markers to distinguish Th1, Th2 and Tfh responses could further strengthen the manuscript.  

Reviewer 2 Report

Comments and Suggestions for Authors

The authors presented the efficacy of a recombinant adenoviral-vectored trivalent (rAd-Tri-Vic) vaccine targeting Victoria-like hemagglutinin (HA) proteins of Influenza B Virus (IBV). Using a computational design approach, the authors selected wildtype IBV sequences to formulate the vaccine, which was tested in mouse models. Key findings include: 1. Strong antibody and T-cell responses post-vaccination. 2. Cross-lineage protection against both B/Victoria-like (B/Vic) and B/Yamagata-like (B/Yam) strains. 3. Durability of immune response up to six months. The study is well-designed, methodologically sound, and provides compelling data supporting the potential of an alternative influenza B vaccination strategy. However, some areas need improvement, particularly regarding statistical analysis, interpretation of cross-lineage protection, and manuscript clarity.

1.      Some statistical analyses are not adequately described, and sample sizes for certain experiments (e.g., ELISpot, HI titers) need more justification.

2.      The study confirms the presence of cross-lineage T cell responses (IFN-γ ELISpot data) but does not fully elucidate the mechanism of cross-protection.

It would be beneficial to assess: B cell response profiling (e.g., ELISA for cross-reactive IgG against B/Yam HA) and maybe Fc-mediated effector functions (ADCC, ADCP assays) to determine if non-neutralizing antibodies play a role.

3.      Some sections are overly dense with technical details (e.g., Results section). It would be good to break down complex sentences for better readability. Provide concise summaries at the end of each major result.

Comments on the Quality of English Language

No comments
